# Ionomer Membranes Produced from Hexaarylbenzene-Based Partially Fluorinated Poly(arylene ether) Blends for Proton Exchange Membrane Fuel Cells

**DOI:** 10.3390/membranes12060582

**Published:** 2022-05-31

**Authors:** Tzu-Sheng Huang, Hsin-Yi Wen, Yi-Yin Chen, Po-Hao Hung, Tung-Li Hsieh, Wen-Yao Huang, Mei-Ying Chang

**Affiliations:** 1Department of Photonics, National Sun Yat-sen University, Kaohsiung 80424, Taiwan; zxp86133@gmail.com (T.-S.H.); rain19950409@gmail.com (Y.-Y.C.); leo3439810@gmail.com (P.-H.H.); 2Department of Green Energy and Environmental Resources, Chang Jung Christian University, Tainan City 71101, Taiwan; hywen@mail.cjcu.edu.tw; 3Department of Chemical and Materials Engineering, National Kaohsiung University of Science and Technology, Kaohsiung 80778, Taiwan; 4General Education Center, Wenzao Ursuline University of Languages, Kaohsiung 80793, Taiwan; tunglihsieh@gmail.com

**Keywords:** blend membranes, poly(arylene ether)s, ionomers, proton exchange membranes, fuel cell, hexaarylbenzene, simulations, cardo structure, partially fluorinated, phase separation

## Abstract

In this study, a series of high molecular weight ionomers of hexaarylbenzene- and fluorene-based poly(arylene ether)s were synthesized conveniently through condensation and post-sulfonation modification. The use a of blending method might increase the stacking density of chains and affect the formation both of interchain and intrachain proton transfer clusters. Multiscale phase separation caused by the dissolution and compatibility differences of blend ionomer in high-boiling-point solvents was examined through analysis and simulations. The blend membranes produced in this study exhibited a high proton conductivity of 206.4 mS cm^−1^ at 80 °C (increased from 182.6 mS cm^−1^ for precursor membranes), excellent thermal resistance (decomposition temperature > 200 °C), and suitable mechanical properties with a tensile strength of 73.8–77.4 MPa. As a proton exchange membrane for fuel cell applications, it exhibits an excellent power efficiency of approximately 1.3 W cm^−2^. Thus, the ionomer membranes have strong potential for use in proton exchange membrane fuel cells and other electrochemical applications.

## 1. Introduction

Fuel cells are energy supply devices whose use has increased in recent years. Because of their low noise level and high energy conversion efficiency, fuel cells have strong potential for use in applications such as transportation equipment and stationary energy devices [1,2,3,4,5,6]. In proton exchange membrane fuel cells (PEMFCs), hydrogen is used as a high-calorific-value fuel to achieve low carbon emission [6,7,8]. The polymer materials used to produce proton exchange membranes (PEMs) are often ionic group–containing ionomers with high proton conductivity (σ_ion_), physical and chemical stability and moisture stability, as well as low electrical conductivity and fuel permeability. These properties of ionomers enable PEMs to have strong performance and high durability [1,3,9,10,11,12]. PEMFCs have operate as an all-solid structure that makes them ideal for transport applications and also have a special polymer electrolyte membrane for conducting protons applications by enhancing electrolyte conductivity. The structure of the perfluorosulfonic acid (PFSA) membranes consists of a fluorinated backbone with a fluorocarbon side chain and possesses characteristics such as excellent ionic conductivity even in anhydrous conditions [13,14,15,16,17,18]. The most common commercial ionomers used for producing PEMs are PFSA-based ionomers, such as Nafion (Dupont) [1,19,20,21,22], Aquivion^®^ (Solvay) [23,24,25,26], Flemion ™ (AGC Chemicals) [27,28,29], and 3M™ Ionomers (3M Company) [1,30,31,32,33]. Numerous studies have developed high-quality PEMs through inexpensive processes involving little pollution [1].

The characteristics of these ionomers make them suitable polymer structures crucial for the development of highly functional PEMs. Polymer blends have attracted interdisciplinary research interest because of their compatibility, simplicity, and low cost [34,35,36,37]. Membranes with suitable separation properties can reduce the risk of uncertainty in terms of development duration.

The polymer architecture resulting from specially functionalized polymers is effective in achieving optimal PEM performance through interactions between hydrophobic and hydrophilic regions [1,38]. This optimization also depends on the characteristics of the polymer chain because it creates ion channels to promote higher proton conductivity and increases dimensional stability because of the stacked arrangement of the polymer backbone or the encapsulation of chemoresistant structures [1,39]. The preparation of blend membranes can gain the benefit of interaction between functionalized polymers or additional support, leading to the formation of new morphologies at the interface of the blend. However, problems are associated with the polymer blending processes. These problems include the balance of mechanical, chemical, thermal, and conductive properties; segregation; and other specific conditions that the new material-specific application performs when blending two or more polymers together, with intersystem compatibility and phase separation being among the key factors [1,40,41].

Several studies have investigated the phase separation of polymer, such as PFSA-containing blends and derivatives of sulfonated poly(arylene)s, through computational simulations [1,42,43,44]. By using dissipative particle dynamics (DPD) simulations, Lu et al. determined the highest compatibility of poly(vinylidene fluoride)-grafted polystyrene sulfonated acid (PVDF-g-PSSA) with sulfonated poly(ether ether ketone) and the changes in cluster patterns caused by different doping ratios. In the study of Lu et al., a high miscibility between the two polymers was shown at a PVDF to PSSA ratio of 50%, at doping ratios of 10% and 20%, and with the addition of (PVDF) to the polymer structure. These studies have also evaluated the smoothness of the miscibility process by simulating states at different times. Scholars have begun to simulate the functioning of fuel cell polymers [42]. Li et al. [45] developed a DPD simulation method based on the coarse-grained method to analyze the three-dimensional (3D) dynamics of polymers at different annealing temperatures and blending ratios. They also used this method to model the host–guest doping region of regular poly(3-hexylthiophene-2,5-diyl) and [6,6]-phenyl-C61-butyric acid methyl ester in organic solar cells and the energy conversion efficiency of the doped cells [39].

This study performed dense sulfonation by using trifluoromethyl (−CF_3_) structures designed for poly(arylene)s to fabricate ready-blended polymer PEMs. The strategy used in this study involved combining a fluorenyl with a cardo structure [46], high torsional orientation, and highly rigid hexaarylbenzene (HAB) derivatives [47] with sulfonated poly(arylene)s. The structures produced through this strategy were expected to exhibit a large proton transport space and appropriate mechanical and thermal stability. To investigate and explain the separation of ionomer blends, it is necessary to have proper conditions in processes. For sufficient ion cluster growth time, a high-boiling-point solvent [dimethyl sulfoxide (DMSO)] system and a fixed processing condition of blade coating were required. Properties of the produced PEMs—namely their mechanical strength, water uptake, dimensional stability, proton conductivity, morphology, and single-cell performance—were characterized, and a detailed, simulation-based morphological investigation of the PEMs was conducted.

## 2. Materials and Methods

### 2.1. General Methods

All reagents and solvents used in this study were purchased from commercial suppliers and used without further purification unless otherwise stated. Dry solvents noted in this study were dried with commercial dehydrating agent and then freshly distilled under an inert atmosphere. All reactions were performed in a repurified N_2_ atmosphere. Diol and difluoro monomers were synthesized by following our previously reported procedures [48,49].The bisphenol monomer 9,9’-bis(4-hydroxyphenyl)fluorene was purchased from TCI Chemical (Tokyo, Japan). One-pot polycondensation reactions were conducted in a fractional distillation glass reactor through water toluene azeotropic distillation to remove moisture from alkaline reaction systems at 100 °C. Subsequently, the reaction temperature was increased to 150 °C. To control the molecular weight range to produce suitable processing, sampling and gel permeation chromatography analysis (GPC) were terminated at a molecular weight of approximately 200 kDa. The steps of the polymer reaction are presented in Figure 1.

### 2.2. General Procedure for Producing Ionomers

At room temperature, 0.5 M chlorosulfuric acid was injected slowly into an 80 mL solution of dichloromethane (DCM) and the prepared polymer (1.6 g). The reaction mixture was stirred for 24–48 h and then precipitated in deionized (DI) water. The polymer precipitate was filtered, washed thoroughly with DI water until the pH became neutral, and then dried in vacuum at 80 °C overnight to obtain the sulfonated polymer (ionomer). The steps involved in the ionomer production reaction are presented in Figure 1.

### 2.3. Preparation of Blend Membranes

Two ionomers, namely s-P12F97B and s-P6F9CB, were produced and dried for 48 h at 80 °C and dissolved separately in 7 wt.% (*w*/*w*) DMSO solvent. A quantity of the s-P12F97B solution was added to the s-P6F9CB solution and stirred at 60 °C for 24 h. Different s-P12F97B:s-P6F9CB mass ratios were used to obtain the BM-1 (90/10) and BM-2 (85/15) solutions. After blade coating, a multistage membrane-drying technique was applied until the solvent was completely removed. The edges were wetted with DI water, and the prepared membrane was then peeled off of the glass plate. Membrane thickness ranged from 25 to 40 μm and was similar to that of Nafion 211 (N211; approximately 25.4 μm), which was used as reference [50].

### 2.4. Measurements

GPC analysis was performed using a Viscotek 270 Max (Houston, TX, USA) device with a refractive index detector as well as a tetrahydrofuran (THF) eluent (a polystyrene standard) at a flow rate of 1.0 mL min^−1^. Thermogravimetric analysis (TGA) was performed using a PerkinElmer Pyris 1 instrument (Shelton, CT, USA) over a temperature range of 50–800 °C and a heating rate of 10 °C min^−1^ under N_2_ atmosphere. Thermomechanical analyses (TMAs) were performed on membrane specimens (length: 10.0 mm; width: 1.0 mm; and thickness: approximately 30.0 μm) by using the PerkinElmer Pyris Diamond TMA device at room temperature.

#### 2.4.1. Oxidative Stability

The produced membranes were weighed and soaked in Fenton’s reagent (3% aqueous H_2_O_2_ solution containing 2.0 ppm ferrous sulfate) at 80 °C for observation after 24 h. Oxidative stability (OS) was evaluated using the change in the weight of the membranes after exposure to Fenton’s reagent.

#### 2.4.2. Water Uptake and Dimensional Stability

The water uptake (WU) of the membranes is gauged by comparing the weights of the dry and the wet membrane samples by means of Equation (1). The dry membrane weight (*W*_dry_) is obtained by vacuum-drying the sample at 80 °C for 24 h immediately before weighing it. The weight of the corresponding membrane in wet conditions (*W*_wet_) is obtained by immersing the membrane sample in DI water at a specified temperature for 24 h, wiping off the surface moisture with filter paper, and then quickly weighing it. The final WU is obtained from the average of three experiments.
(1)$$Water\;uptake\;\left(\mathrm{WU}\right)=\frac{W_{\mathrm{wet}}-W_{\mathrm{dry}}}{W_{\mathrm{dry}}}\times 100\%,$$

The swelling ratio was calculated using Equation (2):(2)$$Through\;plane\;swelling\;ratio\;\left(\Delta A\%\right)=\frac{A_{\mathrm{wet}}-A_{\mathrm{dry}}}{A_{\mathrm{dry}}}\times 100\%$$
where *A*_wet_ and *A*_dry_ are the areas of the wet and dry membranes, respectively.

#### 2.4.3. Ion Exchange Capacity

The ion exchange capacity (IEC) of the membranes was determined through acid–base titration. Prior to titration, each dried membrane was immersed in 1 M HCl (aq.) and 1 M NaCl (aq.) for 24 h to protonate the acid groups and replace the protons of the sulfonic acid groups with Na^+^ ions, respectively. After protonation, the membrane should be washed with water until the waste solution keeps a neutral pH ~7. This should be performed to remove the excess acid on the membrane surface. Then, 1M NaCl can be used to substitute the protons only inside the acid groups of the polymers [51]. Finally, filtrate was titrated using 0.01 M NaOH (aq.), and phenolphthalein was used as an indicator. The final IEC was obtained from the average result of three titrations by using Equation (3):IEC (mmol g^−1^) = (*V*_NaOH_ × *M*_NaOH_)/*W*_dry_(3)
where *V*_NaOH_ and *M*_NaOH_ are the volume and concentration of NaOH (a.q.), respectively, and *W*_dry_ is the weight of the dry membrane.

#### 2.4.4. Proton Conductivity

The membranes’ proton conductivity was measured using a frequency response analyzer (Solartrom 1260, AMETEK, Berwyn, PA, USA) in the in-plane direction with a frequency range of 10 MHz to 100 Hz at a voltage of 100 mV. Conductivity was measured after clamping a 10 × 5 mm^2^ sample between the two platinum electrodes of a conducting cell. The conducting cell was placed in an Espec SH-241 environmental chamber to measure its conductivity at 80 °C and various relative humidity (RH). Proton conductivity (σ_ion_) was then calculated using the following equation: σ_ion_ = *L*/*RA*, where *L* (cm) is the distance between the electrodes, *R* (Ω) is the membrane’s resistance, and *A* (cm^2^) is the cross-sectional area of the sample.

#### 2.4.5. Microstructure Analysis

Transmission electron microscopy (TEM) was performed using a JEM-2100 (high-resolution) TEM instrument (JEOL Ltd., Tokyo, Japan) at an accelerating voltage of 200 kV. After protonation, the acid form of the ionomer blend membranes (BM-X) was dyed by immersion in 1 M AgNO_3_ through overnight immersion. Subsequently, the membranes were washed thoroughly with DI water and vacuum dried at 80 °C for 12 h. The dyed membranes were placed in an enclosure of epoxy resin and ultramicrotome (Leica EM UC7) under cryogenic conditions to obtain samples with a thickness of 70 nm.

Small-angle X-ray scattering (SAXS) measurements using the NanoSTAR U SAXS system (Bruker AXS GmbH, Karlsruhe, Germany). Prior to these measurements, all membrane samples were stained with Ag^+^ ions. VANTEC-2000 detector cooperating the CuKα radiation with a wavelength of λ = 1.542 Å was provided by a Incoatec Microfocus source (IμS) operating at 50 kV and 600 μA. The two-dimensional (2D) scattering data were analyzed, and integrated SAXS intensities were plotted against scattering vector *q*. This vector *q* is a function of angle (*θ*), as presented in Equation (4) [50,52]:(4)$$q=\frac{4\pi Sin\theta }{\lambda }$$

Variable *θ* is the scattering angle. Characteristic separation length d, namely the Bragg spacing, was calculated using Equation (5) [50,52]:(5)$$d=\frac{2\pi }{q}$$

#### 2.4.6. Single-cell Performance

Catalyst ink was prepared by mixing Pt/C (HiSPEC 4000, Alfa Aesar, Haverhill, MA, USA) with a 5.0 wt.% Nafion D520 binder. The ink was fragmented using ultrasonic inkjet printing technologies and then sprayed onto both sides of the prepared catalyst coated membrane. The active surface area was 2.50 × 2.50 cm^2^, with the overall Pt loading being 0.4 mg cm^−2^. A membrane electrode assembly (MEA) was obtained by sandwiching the BM-X membranes between two gas diffusion layers (GDL 24 BC, Hephas Energy Co., Ltd., Hsinchu, Taiwan), and N211 membrane was used as reference. The electrode was supplied with hydrogen at a flow rate of 0.2 L min^−1^ and oxygen at a flow rate of 0.4 L min^−1^. The fabricated cell was activated for 4 h with hydrogen and oxygen at 80 °C and 0.5 V.

### 2.5. Simulation Method

Three-dimensional (3D) modeling of the ionomer molecules was conducted using Materials Studio, and the geometric of the membrane’s optimization was performed using the Forcite simulation suite. A structural model of the blended membranes was constructed on sulfonated-fluorine-containing poly(arylene ether)s (Figure 1). The quantity of sulfonic acid and the number of substitutions in the prepared ionomers corresponded to the experimental IEC. The molecular weight of the repeat unit of ionomers and further parameters were organized in Appendix A. The substitution positions of sulfonic acid group (−SO_3_H) can vary [42,43]; therefore, in this study, three assumptions were made in order to conduct an idealized specific sulfonic acid root substitution position experiment. These assumptions are as follows:The aromatic ring has a low rate of sulfonation substitution because of the deactivation effect in the presence of an electron-withdrawing group (e.g., −CF_3_ and −SO_3_H) [53].Under mild reaction conditions, the substitution reaction on the benzene ring tends to be monosubstitution or substitution at specific positions (e.g., at the 2,7 position on the fluorene ring [46]).The active position has high steric hindrance and a low reaction rate [53].

Although breakthrough substitution reactions (e.g., double substitution on the monophenylene ring) might occur, the assumptions are still valid. In summary, the defined positions of the substitution on the aromatic ring were configured with −SO_3_H as expected and were based on a matched molecular model of the experimental IEC values, and the results were obtained as shown in Figure 1a. The configuration of the sulfonated polymer structure was completed. This setting allows the simulation to match the actual behavior more closely.

The coarse-grained model was used to reduce the calculation time for polymers with many atoms and high molecular weights. Segments containing the −CF_3_ group, which were hydrophobic segments, were named 12F and 6F, while the hydrophilic segments were named 7B-DO and CB-DO (Figure 1). The hydrophilic segments contained seven benzene ring structures and fluorene-based cardo structures, which were easily substituted with sulfonic acid, with origins traced to phenol derivatives. The molecules’ beads were named 7B and CB (Figure 1b).

In all subsequent simulations, a 40 × 40 × 40 nm^3^ cubic grid was simulated at 298 K. The compressibility parameter was 10 kT, the noise parameter was 75.002, and the grid spacing was 1.0 nm. The total simulation time was 300 μs. The simulation comprised 6000 time steps, with each time step (dt) being 50 ns. Two polymers were in the experimental system: the host (s12F97B) and the guest (s6F9CB). Two mixtures with different blending ratios (in terms of weight percentage) of these polymers were examined: BM-1 (90/10) and BM-1 (85/15). Mixtures with blending ratios of 75/25 and 50/50 were also investigated under the same parameters to determine the correlation between polymer blending simulations and experimental morphology.

## 3. Results and Discussion

### 3.1. Synthesis and Characterization of the Monomers and Polymers

The host ionomer, namely s-P12F97B, was obtained through one-pot nucleophilic polycondensation of the difluoro monomer (4,4⁗-Difluoro-3,3⁗-bis(trifluoromethyl)-2″,3″,5″,6″,4,4⁗-difluoro-3,3⁗-bis(trifluoromethyl)-2″,3″,5″,6″-tetra (trifluoromethyl) phenyl-[1,1′:4′,1″:4″,1″:4‴,1⁗-quinquephenyl], 12F9B-DF) and diphenol monomer 2″,3″-Diphenyl-[1,1′:4′,1″:4″,1‴:4‴,1‴-quinquephenyl]-4,4⁗-diol, 7B-DO). The ionomer, namely s-P6F9CB, was produced through polymerization of 4,4⁗-Difluoro-3,3⁗-bistrifluoromethyl-2″,3″,5″,6″-tetraphenyl- [1,1′;4′,1″;4″,1‴;4‴,1⁗]-pentaphenyl (6F9B-DF) and 4,4′-(9-Fluorenylidene)diphenol (CB-DO), as shown in Figure 1. This result is consistent with the results reported in [42,43]. The molecular weights were monitored through GPC. The weight average molecular weights of the host and guest polymers were 201 and 193 kDa, respectively, and their polydispersity index values were 1.8 and 1.9, respectively. Thus, a reduction was achieved in the reduced processing variables required for the solubilization of the polymers in common solvents such as THF, DCM, and chloroform. This indicates that the activation of a fluorine-substituted phenyl group in the neighboring strong electron group enhances the reactivity of a polymer. In addition, the group selected in this study was trifluoromethyl, which contributes to an increase in solubility, chemical stability, and high hydrophobicity that is sufficient to obtain poly(arylene ether)s with a high molecular weight for application to PEMs via the polymerization process [54]. 

Attenuated total reflectance (ATR)–Fourier-transform infrared (-FTIR) spectroscopy experiments are widely used to investigate the structure and interaction of ionomers. The composition of a composite membrane might alter the phase separation of concentrated sulfonated polymers because of their hydrophobic segments and cardo structures. Deformation of ionic clusters affects the topology of the hydrated cluster. Gruger et al. [55] investigated Nafion membranes through ATR-FTIR spectroscopy [1,55]. In this study, weak but identifiable sulfonate peaks (at approximately 1054 cm^−1^) were observed in the ATR-FTIR spectra for both hybrid membranes (Figure 2a). The asymmetric stretching vibrations of these membranes (ν_as_; located at approximately 1100–1400 cm^−1^) are strongly influenced by environmental and matrix characteristics and overlap with the C–F stretching vibrations, which are difficult to locate, in the ATR-FTIR spectra [1,55]. 

This study used the wave number of sulfonic acid (−SO_3_H) to normalize the ATR-FTIR spectra so that the correlation between the molar concentration of sulfonic acid and its functional features could be approximated. The peaks at approximately 1035 and 1010 cm^−1^ in the FTIR spectra correspond to the elongation oscillation of aromatic ethers (Ar-O-Ar) [54]. The absorption features at high wavelengths are affected by the distribution of water clusters generated because of water absorption and the morphology of the microphase in the matrix. Absorption at a high wavenumber produced a partial-collapse-like spectrum. For s-P6F9CB, which was affected by cardo structures, is clearly prominent at 1025 cm^−1^. This result indicates that not only the composition, but also the morphology of the polymer chain differed. Peaks assigned to the vibration of a hydrated cluster were observed at 1500–3800 cm^−1^. This indicated vibration broadband asymmetric stretching vibration (ν_1_, at approximately 3200–3800 cm^−1^) and OH-bending vibration (ν_2_, 1600–1800 cm^−1^) [55]. In this study, conventionally blended aromatic ionomer membranes exhibited a notable difference in OH vibration in an air flow environment and had a wide absorption band for ν_3_ (around 2000–3800 cm^1^). The hydrogen-bonding forces that might arise from a hydrated cluster (H_3_O^+^ (H_2_O)_n_), or from combinations of the sulfonate group (−SO_3_^−^) with different clusters (e.g., (H_2_O)_n_, −CF_3_−, and O=S), require careful experimental verification. For ν_2_, a shoe-like absorption band occurred at approximately 1900 to 1600 cm^−1^ [55]. This band contained the upper band from the toe cap to the tongue, rises at the heel collar, and exhibited a gentle collapse to the heel counter, resulting in a raw identifiable peak, shown in Figure 2b.The tongue peak of BM-1 produced a dramatic plate shift with a bulge at 1703 cm^−1^ that exceeded the tongue center of gravity of s-P6F9CB (1710 cm^−1^). Subsequently, BM-2’s center of gravity returned to 1713 cm^−1^. These results suggest that the hydrated cluster and hydrogenbonding interaction resulted in considerable after-mixing. The absorption feature at 3051 cm^−1^, which corresponded to the C–H elongation of the aromatic ring, was almost covered, leaving only small protrusions on the slope (Figure 2a) but did not affect the intensity of the center of gravity signal at 3417 cm^−1^ with increasing object composition, where the kneecap on the high-wave-number side corresponded to the multiplicity of ν_2_. These results are consistent with the assumptions of Gruger et al. [55]; the causes must be examined by considering nanoscale and mesoscale morphology for applied to the PEMs.

### 3.2. Thermal Stability

The results obtained in the thermogravimetric analysis of the prepared poly(arylene ether) blend membranes (BM-1 and BM-2) are presented in Figure 3 and Table 1. These membranes exhibited similar stability to or higher stability than polymers in other studies [42,43]. For all samples (even under the predrying treatment), a moisture loss of 3 wt% occurred in the temperature range of approximately 100 °C. Although there may have been bound water residue, the rate of moisture loss decreased at 120 °C until the sulfonate acid group was cleaved [1,56]. Subsequently, a major stage of cleavage occurred at >200 °C, which broke the T_d,5%_ norm. In the first-order differential curve, a significant loss can be observed around 200 to 400 °C. This area may be the decomposition of sulfonate sites (SO_3_H to SO_2_ + OH) [1,57,58]. In thermogravimetric curves of reference, this desulfonation was accompanied by decomposition of the ether groups on the side chains from 290 to 400 °C [59,60]. After entering the second plateau, considerable main chain residue was observed after sulfonic acid removal until the second collapse occurred at approximately 540 °C. Before sulfonation, the Td,5% of host and guest poly(arylene ether)s were greater than 600 °C [1,57,58]. The early entries of s-P6F9CB than of s-P12F97B in the second stage of cleavage can be attributed to differences in their main chain characteristics, with the nonbendable multibulk ring structure (structure of a HAB derivative) of s-P12F97B resulting in a higher stiffness than that of s-P6F9CB. To prevent confusion, we calibrated the regression to remove the bound water content and defined T_d,CR_ to evaluate the thermal stability of the membranes. There was no major difference between the s-P12F97B and BM-X membranes, with both membranes failing at approximately 250 °C, which demonstrates that the blending range did not affect the overall cracking. 

### 3.3. Mechanical Properties

The stress–strain curves of the two compositions of BM-X at ambient humidity indicated that the prepared PEMs had appropriate strength characteristics, including Young’s modulus (YM), tensile strength (TS), and elongation at break (EB). Thus, these PEMs can be used in PEMFCs (Figure 4 and Table 1). The membranes exhibited higher YM and TS values than Nafion 211 did (0.28 GPa and 34.2 MPa, respectively) because of their richness in aromatic compounds and rigid nature [48,49,53,61]. IEC indicates the sulfonic acid concentration [−SO_3_H] in standardized matrix. The structure of an HAB derivative with unique stereospecific features can effectively provide a dense sulfonation feature that facilitates conduction [47,48]. Typically, as the IEC of a PEM increases, the resulting hydrogen bonding forces make the membrane more hygroscopic and susceptible to moisture plasticization effects. In this case, moisture acts as a plasticizer [62], and the PEM elongation increases with the IEC of the guest composition (s-P6F9CB), which results in a considerable increase in EB (%). The high EB of BM-2 and the stiffness of its HAB-rich structure resulted in it having a toughness of 81.7%. The high torsional cardo structure in the polymer backbone inhibited chain stacking and ductility, which reduced the effect of the van der Waals force, while the chain entanglement effect resulted in a considerable increase in ductility and the maintenance of the YM value (>0.7 GPa). For ionomers, an increase in IEC is usually accompanied by an increase t in humidity sensitivity. As displayed in Figure 4, increases in the IEC and the quantity of cardo structural components contributed to an increase the EB of BM-1 by 1.5%. The high EB (>80%) of BM-2 might also be attributed to the effects of its rigid matrix. Accordingly, the appropriate component balance for achieving suitable proper ductility and strength must be verified under cell operation.

### 3.4. Oxidative Stability

To predict the chemical stability of the BM-X membranes under extreme operation conditions, they were immersed in Fenton‘s reagent for 1 day at 80 °C [52,63,64], and the results are listed in Table 1. The electron-deficient protection effect caused by trifluoromethyl groups [9,48,65] preserved the morphology of BM-1, which had a rigid structure, and maintained its OS at 79.4%. On the other hand, it also protected the low-OS guest composition in the matrix. However, as the guest concentration increased, the association of the fragile phases subjected the BM-2 membranes to a chain collapse effect, which led to violent disintegration and reduced membrane durability. This also directly limited the allowable amount of high-IEC and high-σ_ion_ objects.

### 3.5. Hydration Behavior

Casting was applied to the ionomer blends (BM-X), with BM-1 and BM-2 consisting of 10 wt.% and 15 wt.% of the host (s-P12F97B) and guest (s-P6F9CB), respectively. As expected, the inhomogeneous composite BM-1 had a relatively isolated phase distribution for the guest; however, BM-2 exhibited a continuous phase distribution for the guest. The hydration behaviors of the blend membranes are presented in Table 2. These membranes had satisfactory uptake (WU) and coverage (approximately 40–70%), and their through-plane swelling (2D change rate, ΔA (%)) was less than 45%. The rigid structure and superhydrophobic characteristic of the trifluoromethyl groups on the polymer backbone resulted in the dimensional stability of BM-X being maintained, which resulted in a moderate increase in ΔA (%) under a high concentration of a high-IEC guest polymer (51.2% at 80 °C) and suppression of its sensitivity to moisture. In this study, the guest had a sulfonated HAB-derived structure, and the localized hydrophilic architecture formed by it and with the torsional freedom of its cardo structure resulted in a dramatic change in its WU. The increase in the WU of the guest was an order of magnitude higher than that of the blend membrane (277.9%). Even so, they were water-miscible at 80 °C. Thus, maintaining the dimensional stability of the substrate polymer’s architecture is crucial. In addition, the efficiency of a cell can be increased by maintaining a nonuniform distribution of high-IEC phases. 

### 3.6. Proton Conductivity

Proton conductivity σ_ion_ as function of the RH (%) at 80 °C is the main factor that determines whether BM membranes can be used as PEMs. The variations in the proton conductivity of the prepared BM membranes at different RH levels were determined (Table 2 and Figure 5). The collected data indicated that the σ_ion_ of all membranes was correlated with RH% and showed a significant increase at an RH of 80%. The BM-1 and BM-2 membranes exhibited maximum σ_ion_ values of 191.2 ± 12.9 and 206.4 ± 10.2 mS/cm^−1^, respectively. These membranes had higher σ_ion_ values than N211 (110.0 mS/cm^−1^) did. Although s-P6F9CB exhibited the highest σ_ion_ value (277.9 ± 18.4 mS/cm^−1^) in this study, it had poor oxidation stability. As expected, the σ_ion_ value increased with an increase of the guest concentration at a high IEC. At a low RH, the σ_ion_ value did not increase considerably with an increase in the guest concentration, probably because the concentration of sulfonic acid at the pool hydrated level did not vary substantially; thus, the surface mechanism of mass transfer did not change considerably with the guest concentration. As the ambient RH increased, the in WU increased, and the Grotthuss and vehicular mechanisms became increasingly dominant [1,66]. The BM-X membranes had higher σ_ion_ values than the reference (122.6 mS/cm^−1^) at an RH of 80%. The BM-1 membrane exhibited an increase in water content with increase in WU variation; however, it did not exhibit the highest σ_ion_ value in this study, which might be attributable to the polymer architecture of the blending system. This, coupled with the high dimensional stability of BM-1 membranes, indicates that BM-1 membrane is expected to be an excellent PEM.

### 3.7. Microstructure Analysis

To observe the microphase separation pattern of the blend ionomer (BM-X) membranes, the membranes were embedded and sliced, and this section was analyzed through TEM and SAXS. The cross-sectional TEM images are presented in Figure 6. All membranes exhibited a clear microphase separation pattern. The distributions of the dark regions in these figures differ considerably, which indicates that the complex composition strongly affected ionic cluster aggregation. The images also reveal the original distribution of the hydrophilic and hydrophobic regions. The shape, density, location, and relative grayscale level of these clusters provide a basis for determining the topology of the hydrophilic region and are often considered the main springboard for proton transport [1]. Accordingly, the relationships of the formation of the blend ionomer with proton conductivity and other basic properties were investigated through morphological analysis of the cluster distribution. As shown in Figure 6a,b, the ionic cluster distributions of s-P12F97B and s-P6F9CB membranes occurred both in independent and dense island-like distributions (diameter less than 10 nm), the latter being more densely due to the high torsional angle of their cardo structure and relatively high IEC feature. Notably, in the composite BM-1 containing guest of 5 wt.%, mesoscale appears as vesicle-like clusters with high aggregation density (high hydrophilic zone) in its cluster distribution network (long axis of approximately 610 nm and short axis of approximately 275 nm). Mesoscale had more variable spacing between clusters, around 330 nm to 1100 nm, and the spaces between clusters (low hydrophilic and hydrophobic regions) still had a lower density of clusters. The depth of the linked TEM image, where the dark shadows appear as heavy atoms coalescing, also implies excellent IEC, indicating that this region can act as a node for proton acceleration during fuel cell operation. In this study, the clusters formed by the concentrated clusters were primarily in the acceleration region. Together with the coherent hydrophobic region supporting the structural backbone of the membrane layer, this is expected to promote the PEMs with good balance properties. At BM-2 (85/15), although island-like clusters (8.5 to 23 nm in diameter) can still be observed, the morphology tends to be irregular and the mesoscale morphology extends to ribbon fragmentation, even if the boundaries were gradually blurred (Figure 6d). Ion clusters expand as membranes absorb water to form the main channel for proton transport [1,63,64,67,68]. Because BM-1 had a sparse cluster distribution, this membrane was expected to contain a complex backbone or microchannel system for proton transportation. The occurrence of this morphology is attributed to the interaction between s-P12F97B; a rigid-plate-containing backbone; and s-P6F9CB, which had a high torsional angle. Normally miscible polymer chains are expected to interweave into a homogeneous substrate when two polymers are homogeneously miscible in a solvent system. However, other mechanisms are preferred for the formation of PEMs. Because of the different environmental suitability of different structures, deviations or rearrangements during the membrane formation and drying processes can prevent a high concentration of sulfonic acid chains from forming hydrophilic domains, which are also known as “ionic clusters”. Immediately thereafter, proton propagation paths should pass or leap in this high-[−SO_3_H]-enriched plate. The clusters displayed in the TEM images expanded under corresponding transitions to form partially miscible proton propagation channels after moisture absorption.

To identify the distribution and shape of the ionic cluster morphologies, the blend membranes were subjected to SAXS analysis (Figure 7). The characteristic separation lengths *d* of these ionomers were calculated and included in Figure 7, where *d* was the distance between the ion-rich domains in the hydrophobic polymer-rich domains of the ionomer [50,69,70]. BM-1 and BM-2 exhibited maximum contrast intensity (*q*_max_ values at 0.52 and 0.44 nm^−1^, respectively; also ionomer peaks) [50,69,70]. The division of BM-X membranes occurred in the high-q-value direction of shoulder-like hillside, which differs from the Gaussian distribution peak pattern exhibited by the guest polymer with a high torsional angle of its cardo structure. As expected, BM-1 shows a similar distribution to BM-2; however, the distribution of BM-1 was closer to that of the host matrix (s-P12F97B: q_max_ = 0.23 Å^−1^ and *d* = 2.92 nm) than to that of BM-2. BM-1 and BM-2 had intercluster distances d of 2.89 and 2.72 nm, respectively. These values are similar to that obtained for the reference (average *d* of approximately 2.4 nm) [50]. The presence of a cardo structure with a high torsional orientation caused the polymer chain of the highly sulfonated guest to facilitate embedding of the hydrophilic chain segments of the ionomer so that high-density ionic clusters could form. The range of BM-X covers the smaller intercluster distance (*d* = 1.67 nm) region of the guest (s-P6F9CB) self-assembly, which may also reveal a cluster distribution with multiple species; the TEM images support this finding. In the hybrid polymer system, a multiphenyl structure with high torsion and a high IEC might be able to support the generation of dense channels and the carrying capacity of high-proton-transport fluxes without completely minimizing channel interclustering. This does not result in a less-pronounced phase separation morphology with high torsional and high IEC combination in the embedding [50,69,70]. Thus, suitable mechanical properties can be maintained without excessive swelling deformation under a high proton supply and WU. Among them, the longer continuous hydrophobic section and rigid structure were favorable for maintaining proper water-soluble behavior in higher-proton-state space [50,69,70], which was expected to maintain good mechanical properties during cell operation.

To understand the relationship between polymer architecture and morphological characteristics, mesoscopic simulation was performed through coarse-grained modeling. We compared the morphological evolution processes of the 90/10 and 85/15 blends (Figure 8). The blends exhibited solitary islands and worm-like structures, respectively. For the 90/10 blend, the isogenous surface deformed because of the force application and split and coalesced into homogeneous and independent (solitary islands) ionomer clusters (size of approximately 10 nm; Figure 8a). The phase in the periodic 3D cell transformed from a metastable phase to a steady-state phase. For the 85/15 blend, dispersed spheres gradually linked together during the coagulation process and formed a large homogeneous distribution (Figure 8b). Although a connected network structure was not evident, the simulation suggested that considerable microphase separation occurred in the two examined material systems, which might be attributed to the cardo-structure on the guest polymer’s main chain. The relatively high IEC of the client was also influenced by the separation behavior driven by the hydrophilic–hydrophobic difference, which contributed to morphological changes. The linking fragments or islands allowed the formation of the ion transmission channel to be predicted.

Further insight into the morphologies was obtained from the isopycnic plots indicating the density distribution of the side chains of phenyl sulfonic acid in four ionomer blends: 90/10, 85/15, 75/25, and 50/50. In Figure 8, the isopycnic surfaces of all blends are presented as colored 3D contours in a periodic box. In the figure, the red in the center of the cluster nucleus indicates high density, and the sequence from orange to yellow and green to blue (from the nucleus to the shell layer) indicates a gradual decrease in density. However, as the quantity of the high-hydrophilicity (high-IEC) cardo ionomer (s-P6F9CB) increased, the isopycnic surface clusters coalesced and topologized, producing distinct worm-like structures, which indicate increasing density (Figure 8b,c). The simulation indicated that the state in the composition interval is not fully energy-stabilized or geometrically stable, which indirectly suggests that the experimental process for preparing membranes might be highly sensitive to process factors that cause considerable polymer rearrangement and agglomeration because of segregation. By the time the composition reached 50/50 (Figure 8b,d), a well-defined mesh-like structure had formed and retained the island-like independent cluster feature, which is useful for the prediction of proton transport. Thus, the simulation results do not directly link to the mesoscopic cluster morphology, which includes some sparsely distributed cluster regions and irregular plates formed through particle aggregation on the hydrophilic side (−SO^−^_3_H). Because of differences in the process factor and solubility during the preparation of membranes, differences arose in the compositions of regions containing mesoscopic structures. The simulation results indicate that mesoscale island-like clusters (visible in the TEM images) and high-density cluster villages formed in some regions.

### 3.8. Fuel Cell Performance

Partially fluorinated poly(arylene ether)-based MEAs were created using a mixture of s-P12F97B and s-P6F9CB. Then, catalyst-coated membranes were created by spraying Pt/C catalyst on alternate paths. The performance of the cells is illustrated in Figure 9. For cells with cathode gas feeds of oxygen, the maximum power densities of BM-1, and BM-2, and N211 (the reference) were 1.30, 0.65, and 1.25 W cm^−2^, respectively, compared with 1.25 W cm^−2^ for the PFSA reference (N211). At 0.6 V, the current density of BM-1 was similar to that of N211 (1300 mA cm^−2^). At a high current density (>3000 mA cm^−2^), BM-1 membrane showed significant concentration polarization, which resulted in a sharp decrease in cell voltage. This decrease was probably limited by σ_ion_ or the proton flux, as in s-P12F97B (approximately 180 mS cm^−1^), whose cell performance was marginally inferior to that of N211. The cell performance worsened considerably in the kinetic region, which is attributed to the tortuous conduction path and solution in the PEM. BM-2 exhibited visible wrinkles and breakage after battery operation because of the dynamic balance of gas and water management (Appendix A). The balance of PEM characteristics directly affects cell performance.

## 4. Conclusions

The results indicate that polymer structures affect their proton transportation profiles, but that proton transportation can still be induced at the micron or nanoscale through composition changes and fabrication processes. The hybrid polymer system developed in this study enabled structural changes, which resulted in changes in properties. The HAB plate structures and curled cardo structures of the guest polymer were aggregated, embedded, and aligned so that ion clusters formed dense localized clusters (with a size of approximately a few μm) at mesoscopic scales, which are particularly evident in the BM-1 (90/10). The ATR-FTIR data indicated morphological differences between hydrated clusters. In addition to indicating the total quantity of protons that must migrate when high-IEC objects are involved, the SAXS results suggest that increasing the size of the proton channel (from 1.67 to 2.92 nm) might increase also raise the upper limit of the proton flux. Simulations were conducted to analyze the effects of variations in the guest polymer composition on the phase separation. The optimal dispersion of 10% led to a satisfactory compatibility variation of 15–25% for BM-1 and BM-2. Although guest phases with a higher IEC value provided more sources of H^+^ ions and enabled a higher σ_ion_ value, an increase in the guest polymer’s concentration led to the linkage and ductility of the phases. When mixed in equal proportions, the dendritic structure has negatively affected mechanical properties and dimensional stability. Because of differences in solubility and compatibility in the hydrophobic intrachain and interchain segments of polymers, segregation during membrane preparation may have aggravated local composition changes and made the membrane function unbalanced. This phenomenon and the effect of the polymer’s OS resulted in a considerable decrease in the cell performance of BM-2. However, compositional changes quickly improved properties of the prepared PEMs, and acceptable PEMFC performance was achieved (power efficiency of approximately 11.30 W cm^−2^ for BM-1). The unique compatibility and self-assembly properties of the materials were calculated DPD simulations, and the results were used as a basis to develop membranes with balanced proton transport and mechanical properties.

## Data Availability

Data are available in a publicly accessible repository.

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
