# Peer review of "Ionomer Membranes Produced from Hexaarylbenzene-Based Partially Fluorinated Poly(arylene ether) Blends for Proton Exchange Membrane Fuel Cells"

_membranes, 2022, doi:10.3390/membranes12060582_

Round 1
Reviewer 1 Report
In this manuscript, various membranes composed of hexaarylbenzene and fluorene based poly(arylene ether)s were prepared for PEMFC applications. All blend membranes demonstrate a significant increase in proton conductivity compared to the precursor membranes at 80°C, and they were still retaining excellent thermal resistance and admirable mechanical properties. During the PEMFC single-cell test, it exhibits excellent power efficiency (about 1.30 W cm-2), which may be the potential candidate for the practical application in PEMFCs.
I may give a minor revision due to further improvements that are needed by addressing my comments made below.
- Abstract and Introduction
- There are many grammar problems and spelling errors. It is suggested to greatly improve the language. For example:
Line 119, ‘and solutions were prepared separately with DMSO solvent in the ratio of 7.0 wt% (w/w on the basis of the solvent), respectively.’
Throughout the main text, [ref] should be removed and suitable references should be added.
- For the Keywords, ‘partially fluorinated’ and ‘phase separation’ should be added to attract a broader readership.
- Line 38, ‘The common PEM material on the market is per-fluorosulfonic acid (PFSA)’. It should be added here to explain why this kind of membrane has achieved a wide range of commercial applications, and what is its structure? What characteristics does this structure provide for the material to make it suitable for PEMFC applications?
For example, the PFSA membranes contain the hydrophobic fluorocarbon chains and the hydrophilic sulfonic acid groups, and the proton transportation channels are easily formed due to the phase separation between the hydrophobic and hydrophilic domains, when the membrane is hydrated [Ionics 25.9 (2019): 4219-4229; Physical Chemistry Chemical Physics 19.24 (2017): 16013-16022].
- Line 86, the sentence ‘In order to investigate and explain the biased morphology of the PEM composition and its relationship with the characters in the phase transfer process’ seems not completed yet (only half-sentence is shown). In addition, the latter sentence, ‘A high boiling point solvent (dimethyl sulfoxide, DMSO) system gives a fixed processing condition for blade coating’.
- Materials and Methods
- Line 122, ‘The mixture was blended by mixing different mass ratios of s-P12F97B and s-P6F9CB, defined as BM-1 (90/10) and BM-2 (85/15).’Why are only two ratios selected? How about 80/20 and 70/30? Why is the ratio not further changed? The latter calculated (75/25 and 50/50) is too far away from the two ratios adopted, and they are not experimental work as well.
- Line 152, ‘Prior to the test, each dried membrane was then immersed in 1.0 M HCl (aq.) and 1.0 M NaCl (aq.) for 24 h to protonate the acid groups and replace the protons of the sulfonic acid groups with Na+ ions, respectively.’After protonation, the membrane should be washed with water until the waste solution keeps a neutral pH~7. This should be done to remove the excess acid on the membrane surface. And then 1M NaCl can be used to substitute the protons only inside the acid groups of the polymers [Electrochimica Acta 378 (2021): 138133].
- When the MEA was prepared, is the hot-pressing technique used? If it is used, how about the treatment details (time, temperature, and pressure)? And how about the back pressure during the PEMFC single-cell measurement?
- Results & Discussion
- Page 8, the TGA result of Nafion should also be added for comparison since in the single-cell the membrane is also compared with N211 as reference. The same applies to the stress-strain curve, Table 2, and oxidative stability.
- For the TEM images, they are surface or cross-section images that should be clarified. And why cross-section morphology is not shown? This is important to verify whether the membrane is homogenous or not.
- The discussion for the single-cell part should be further explained and compared. Now it seems only focuses on the power density.
Author Response
May 27, 2022
Dear Editors:
Please find enclosed our manuscript titled “Hexaarylbenzene Based Partially Fluorinated Poly(arylene ether)s Blend Ionomer Membranes for PEMFC” which we would like to submit for publication as a research article in Membranes.
In this study, a series of high molecular weight ionomers of hexaarylbenzene and fluorene based poly(arylene ether)s were synthesized conveniently through condensation and post-sulfonation modification. The use a of blending method might increase the stacking density of chains and affect the formation both of interchain- and intrachain proton transfer clusters. The mulitscale phase separation caused by the dissolution and compatibility differences of blend ionomer in high boiling-point solvents was examined through analysis and simulations. The blend membranes produced in this study exhibited a high proton conductivity of 206.4 mS cm-1 at 80°C (increased from 182.6 mS cm−1 for precursor membranes); excellent thermal resistance [decomposition temperature > 200℃]; and suitable mechanical properties with a tensile strength of 73.8–77.4 MPa. As a proton exchange membrane for fuel cell applications it exhibits an excellent power efficiency of approximately 1.3 W cm-2. Thus, the ionomer membranes have strong potential for use in proton exchange membrane fuel cells and other electrochemical applications.
We believe these findings will be of great interest to the readers of your journal. As a premier international journal devoted to the rapid dissemination of research in the field of latest developments in all engineering related fields including communication, biomedical, optical and device technologies, Membranes is the perfect platform for us to share these results with the international research community.
We confirm that this manuscript has not been published elsewhere and is not under consideration by another journal. All authors have approved the manuscript and agree with submission to Membranes. The authors have no conflicts of interest to declare.
We look forward to hearing from you at your earliest convenience.
Sincerely,
Mei-Ying Chang & Wen-Yao Huang
Department of Photonics, National Sun Yat-Sen University, Kaohsiung 804, Taiwan
E-mail: mychang01@mail.nsysu.edu.tw (M.Y.C.); wyhuang@mail.nsysu.edu.tw (W.Y.H.)

Reviewer 2 Report
Dear Author,
The article "Hexaarylbenzene Based Partially Fluorinated Poly(arylene ether)s Blend Ionomer Membranes for PEMFC" is interesting. Here are my comments:
(1) The author normalizes the FTIR signal to calculate and show the other functionality in determining the functional group. Is normalization done on the basis of molar concentration to get the spectra or by weight percent?
(2) In line 315, the author forgot to put the reference number of the article. Also, the Author can also take 1st derivative to show the transitions and decomposition point. How do you determine the sulfonate cleavage and at what temperature it takes place?
(3) SAXS results were not clearly explained by the author. Does the Author fit the data and if yes which model was used to fit the data? How only one q peak suggests about the channel. It only suggests phase separation with ionomer peak. Phenomenal work has been done by Timothy long and segalman on it.
(4) How SAXS data is related to observed mechanical properties. How morphological features affect their overall properties should be explained. Also, please cite the necessary references thereof.
Author Response

(The authors gave the same response as above.)
